# Spectral interferometry with waveform-dependent relativistic high-order harmonics from plasma surfaces

Dmitrii Kormin[1,2], Antonin Borot[1], Guangjin Ma [3,4], William Dallari[1], Boris Bergues[1,2], Márk Aladi[5], István B. Földes[5] & Laszlo Veisz [1,6]

The interaction of ultra-intense laser pulses with matter opened the way to generate the shortest light pulses available nowadays in the attosecond regime. Ionized solid surfaces, also called plasma mirrors, are promising tools to enhance the potential of attosecond sources in terms of photon energy, photon number and duration especially at relativistic laser intensities. Although the production of isolated attosecond pulses and the understanding of the underlying interactions represent a fundamental step towards the realization of such sources, these are challenging and have not yet been demonstrated. Here, we present laser-waveform-dependent high-order harmonic radiation in the extreme ultraviolet spectral range supporting well-isolated attosecond pulses, and utilize spectral interferometry to understand its relativistic generation mechanism. This unique interpretation of the measured spectra provides access to unrevealed temporal and spatial properties such as spectral phase difference between attosecond pulses and field-driven plasma surface motion during the process.

[1] Max-Planck-Institut für Quantenoptik, Hans-Kopfermann Straße 1, 85748 Garching, Germany. [2] Ludwig-Maximilian-Universität München, Am Coulombwall 1, 85748 Garching, Germany. [3] School of Electronics Engineering and Computer Science, Peking University, 100871 Beijing, China. [4] Shenzhen SoC Key Laboratory, PKU-HKUST Shenzhen-Hong Kong Institution, 518057 Shenzhen, China. [5] Wigner Research Centre for Physics, Hungarian Academy of Sciences, Budapest, Hungary. [6] Department of Physics, Umeå University, SE-90187 Umeå, Sweden. These authors contributed equally: Dmitrii Kormin, Antonin Borot, Guangjin Ma. Correspondence and requests for materials should be addressed to L.V. (email: laszlo.veisz@umu.se)

The investigation and control of ultrafast physical processes in nature call for ever shorter flashes of light. The present state of the art is high-order harmonic generation (HHG) from gas medium[1], providing extreme ultraviolet (XUV) and X-ray pulses with durations in the attosecond regime, the temporal scale of electron motion in atoms and molecules. An alternative is represented by HHG from plasma mirrors (PMs)[2,3]. Its main advantage over gas HHG is the potential to utilize lasers with ultrahigh peak power and thus produce bright attosecond light pulses with orders of magnitude of higher energy and shorter wavelength. Such high-energy attosecond light sources will satisfy the challenging needs of attosecond XUV nonlinear optics for XUV-pump–XUV-probe experiments[4]. There have been many advances in the past two decades in understanding and developing attosecond light sources using PMs. So far, three distinct mechanisms of HHG from PMs have been identified. Coherent wake emission (CWE)[5] below relativistic laser intensities and relativistically oscillating mirror (ROM)[6–9] are compared in the experiment and clearly differentiated from each other[10,11]. The ROM model predicts a power law decay of the harmonic spectrum with the harmonic order with an exponent of −8/3. However, there exist different alternative models describing this last regime and providing different exponents[12–15]. Recently, a third mechanism termed coherent synchrotron emission[16] or relativistic electronic spring model[17–19] was proposed and experimentally identified with its characteristic spectral signatures[20]. CWE provides temporally coherent and synchronized XUV harmonics resulting in attosecond temporal bunching[21,22] but with a chirp related to sub-laser-cycle dynamics of plasma electrons[23], which makes it less attractive for applications. The good spatial coherence of the laser is preserved during the CWE process[24]. To date, ROM was observed up to multi-keV[25,26] energy in good agreement with theoretical predictions[9]. It is shown to provide well beamed radiation[27,28] and diffraction-limited focusing[29,30]. The motion of the plasma surface during the interaction (denting) influences the harmonic beam divergence[31] and reduces the coherence length of ROM harmonics[32]. The generation condition is controlled by the plasma scale length to optimize the harmonic yield[33–36] and to change the spacing between pulses in an attosecond pulse train (attotrain)[37]. Furthermore, utilizing similar control, harmonics provide spatial information about structured plasma surfaces via ptychographic imaging[38]. Engineering the instantaneous waveform with two-color driving field control is also proposed and used to enhance high-harmonic yield[39,40].

For future applications of these light sources, isolated attosecond pulses (APs) are superior to an attotrain. For this reason, different polarization gating techniques were proposed for many-cycle laser pulses[41–43], which have large intensity losses. Owing to the advent of intense few-cycle laser drivers with high contrast[44–47], however, direct isolation is within reach through intensity gating[48–50]. In these cases[49], an attotrain is produced containing a well isolated AP with very high isolation degree (defined as main-to-side pulse temporal intensity ratio), which still presents apparent modulation in its corresponding spectrum. A natural property determining the waveform of these few-cycle laser pulses is the carrier-envelope phase (CEP) $\phi_{CEP}$ (see Methods). CEP dependence and control of CWE harmonics was already demonstrated[51] and isolation was reached with tilting the pulse front of the intense few-cycle driver laser and thus producing a manifold of isolated APs propagating in different directions[52]. Although high harmonics from few-cycle driver are expected to be waveform dependent[49], neither this CEP dependence nor isolated APs have been achieved in the relativistic generation regime before.

In this article, we present laser-field-dependent emission of high-order harmonics from a ROM driven by a two-cycle laser. For certain CEP values, this harmonic radiation supports a strongly isolated AP. An analysis based on spectral interferometry (SI) reveals the spectral phase difference between the pulses and the spectrum of the individual pulses in the attotrain. This information allows determining the temporal spacing and relative contrast between individual APs in a few-pulse attotrain as well as the denting of the reflecting solid-density PM[31]. The simultaneous measurement of the laser waveform provides clear demonstration of the CEP dependence of the underlying relativistic laser–plasma interaction.

## Results

**Experiment description.** The experimental set-up is shown in Fig. 1. The multi-terawatt (multi-TW) peak power Light Wave Synthesizer 20 (LWS-20)[44] was used as a laser source with a low-intensity PM for contrast improvement[45] by a factor of 300. It provided a pulse energy of 40 mJ on-target after losses, which was focused with an off-axis parabolic (OAP) mirror to $d_{FWHM} = 1.3$ μm, in an angle of incidence $\alpha_{inc} = 55°$, and p-polarization. Focus size and quality were controlled using a microscope objective, which can replace the target and image the attenuated beam focus to a charge-coupled device (MO+CCD). Chirp-scan technique with a home-made dazscope[53] device was used to measure the on-target compression and optimize it with the help of the acousto-optic shaper (Dazzler) in the laser system. In combination with an almost octave-spanning spectrum, which is slightly red-shifted by the PM optics, it led to two-cycle laser pulses with slightly modified parameters (central wavelength

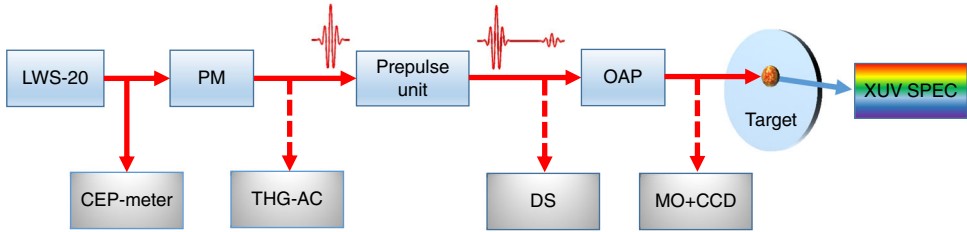

**Fig. 1** Experimental set-up. LWS-20 delivers relativistic two-cycle laser pulses. A small portion of the beam is used in a stereo-ATI phasemeter to measure the CEP for every shot. The contrast of the main pulse is improved by 2.5 orders of magnitude by a plasma mirror (PM) and characterized by a third-order autocorrelator (THG-AC). A controllable prepulse is produced in the prepulse unit. Afterwards, the on-target pulse duration is measured and optimized with a home-made dazscope (DS) in combination with a Dazzler in the system. An off-axis parabolic mirror (OAP) tightly focuses the beam on the fused silica target that is translated after every shot. The generated XUV emission is reflected to a flat-field spectrometer where it is spatially separated from the fundamental beam. Focus quality is checked and optimized with the attenuated beam by replacing the target with a microscope objective (MO) and imaging it to a CCD

of the laser $\lambda_L = 765$ nm, intensity full-width-half-maximum (FWHM) duration $\tau_{FWHM} = 5.1$ fs) with ultra-relativistic peak intensity of $I_L = 1.3 \times 10^{20}$ W cm$^{-2}$ in focus, which corresponds to $a_L = 7$ normalized vector potential defined as $a_L = \sqrt{I_L(\text{Wcm}^{-2})\lambda_L^2(\mu m)/1.38 \times 10^{18}}$. Detailed information about the laser and the pulse characterization can be found in Methods and in ref. [44].

The laser was focused to a fused silica target and the created relativistic PM reflected the driver pulse together with generated XUV harmonic emission toward a flat-field imaging spectrometer equipped with a micro-channel plate and a CCD (XUV SPEC). Each recorded XUV spectrum was tagged with the relative CEP of the corresponding laser pulse, measured with a single-shot stereo above-threshold-ionization phasemeter[54] (CEP-meter).

The XUV generation process is strongly dependent on the preplasma extension at the arrival of the main pulse[33–35], that is the plasma scale length $L$ (see Methods). Therefore, it significantly increases the requirements for the temporal laser contrast, which was characterized by a home-made third-order autocorrelator[55] (THG-AC) when the PM was implemented. The contrast was estimated to be $10^{-19}$ beyond 30 ps and $3 \times 10^{-8}$ at 1.5 ps before the main pulse[44]. A tailored prepulse with adjustable delay was introduced (see Methods), which together with the PM provided a complete control over the plasma scale length for optimization of the XUV generation efficiency and further investigation of the light–plasma interaction. The scale length for the different prepulses was estimated with the hydrodynamic code MEDUSA[56] (see Methods).

**Waveform-dependent XUV spectra.** High harmonics were generated up to 80 eV photon energy on plasma surfaces with optimal scale length. The harmonic spectra were fitted for many shots and found to scale with the harmonics frequency in a power law $I_\omega \sim \omega^{-2.55\pm0.21}$ for $L/\lambda_L \approx 0.13$, close to the theoretically expected ROM scaling[9] (dashed line in Fig. 2a shows a typical fit). For more details on the fit results, see Supplementary Table 1. We thus conclude that in the observed spectral range of 16–100 eV XUV emission was dominantly generated via the ROM mechanism. Measured spectra for two different CEPs with $\pi$ difference (Fig. 2a) had harmonics with photon energies that did not correspond to the integer multiples of the laser central frequency ($\hbar\omega_0 = 1.62$ eV) and were shifted relatively to each other by $\hbar\omega_0/2$. In general, the photon energy of harmonics varied with CEP for different laser shots. Sorting the measured spectra according to the corresponding laser pulse $\phi_{CEP}$ reveals continuous shift of the photon energy of harmonics.

Figure 2b displays normalized and contrast enhanced (see Methods) CEP-sorted spectra measured with a prepulse delay of $\tau_{pp} = 1.67$ ps corresponding to a plasma scale length of $L/\lambda_L \approx 0.13$ (see Methods). A set of one-dimensional (1D) particle-in-cell (PIC) simulations have been performed (see Methods) for the same laser intensity, a series of $\phi_{CEP}$ values, and several different plasma scale lengths. Figure 2c shows the particular case $L/\lambda_L = 0.13$, which is in reasonable agreement simultaneously with the experimental data in Fig. 2b as well as the later evaluation in Fig. 5. Supplementary Fig. 1 shows the simulated CEP-sorted spectra for different scale lengths and indicates that if only the results in Fig. 2 are compared then $L/\lambda_L = 0.1$ shows a better agreement. The absolute CEP in experiments was determined by comparing simulations and experiment (see Methods). Simulations predicted an almost linear shift in the photon energy of the $n$th harmonic within ($\hbar\omega_n - \hbar\omega_{n+1}$) with CEP in the whole energy range. However, this shift increases with the photon energy and its dependence becomes more complex. The same trend is observed in measured spectra: the lowest harmonics have linear shift with

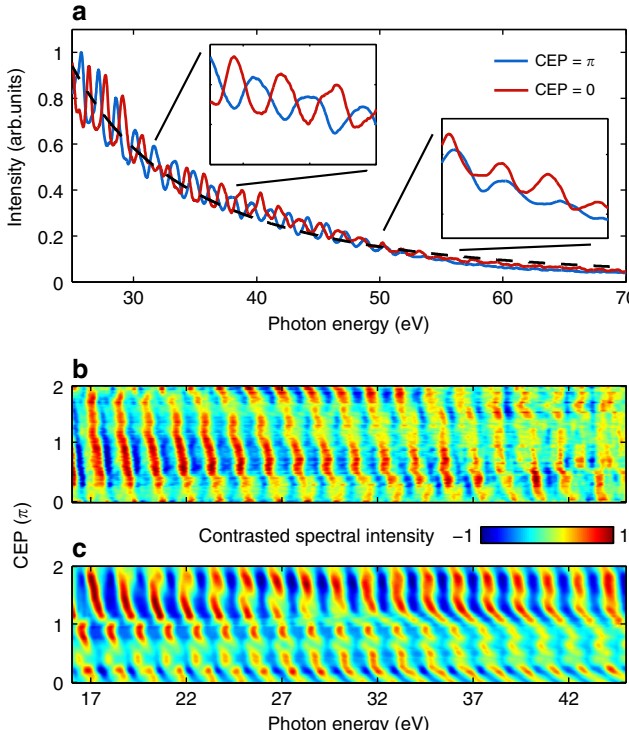

**Fig. 2** CEP-dependent XUV emission. **a** Example of measured XUV spectra generated by the laser pulses with $\pi$-shifted $\phi_{CEP}$. Two insets demonstrate harmonics photon energy and shape difference in spectral regions 30–35 and 50–55 eV, respectively. The dashed line is a power law fit with an exponent of $-2.62$. **b** CEP-sorted experimental XUV spectra for 71 shots for a prepulse delay $\tau_{pp} = 1.67$ ps ($L/\lambda_L \approx 0.13$). **c** Corresponding CEP dependence obtained from PIC simulations with a scale length of $L/\lambda_L = 0.13$. Additional contrast enhancement procedure (see Methods) was applied to measured and simulated spectra for better visualization of harmonics photon energy. A vertical running average smoothing of the data in **b**, **c** was applied within the CEP measurement error range of ±210 mrad

$\phi_{CEP}$ while the highest ones become more disordered. An increase of the plasma scale length also leads to stronger and more complex CEP dependence of the harmonics in simulations, which was experimentally confirmed by increasing the prepulse delay. These results clearly demonstrate a relativistic waveform-dependent interaction.

Such an effect can be explained by treating the obtained spectra as an interference of several APs. XUV pulses appear as a result of the motion of relativistically accelerated electrons driven by the laser electric field. Different field shapes, that is CEPs, therefore lead to different temporal structures of the generated attotrain. Understanding this dependence represents an important step toward the control of the temporal structure of attotrains and even the routine generation of isolated APs with CEP-stabilized few-cycle lasers.

**Spectral interference.** The individual intensity of generated APs is not only connected to the field of the driver laser at the corresponding cycle but also depends on the scale length[49]. According to our simulations, for $\phi_{CEP} \approx 0$ (and $L/\lambda_L \approx 0.25$) two almost equivalent APs are produced, that is they have much higher intensity and maximum photon energy than other pulses. Correspondingly, above a certain photon energy the generated harmonic spectrum can be considered as the interference of only these two strongest APs (see, for example, Fig. 3a). Therefore, SI[57]

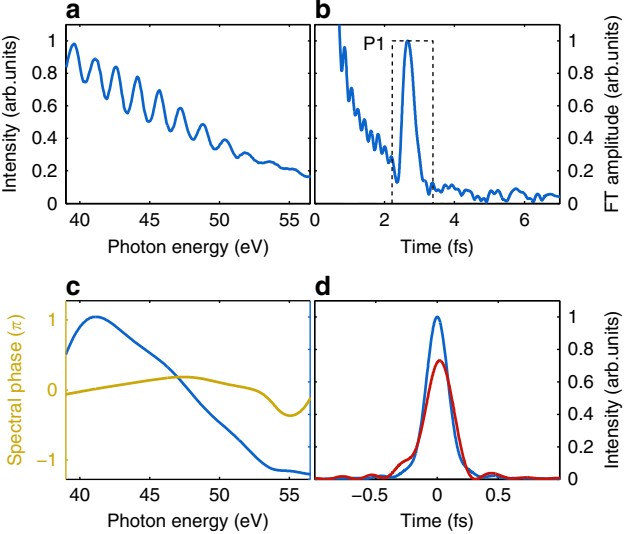

**Fig. 3** Spectral interferometry with two APs. **a** Measured XUV spectrum with $\phi_{CEP} \approx 0$ and $L/\lambda_L \approx 0.25$ in a selected spectral range. **b** Corresponding FT using the 39–57 eV range. The peak P1 represents the interference of the two involved APs. Dashed line shows temporal gate function chosen for the subsequent inverse FT. **c** Reconstructed combination of two AP spectra $S_{P1} = \sqrt{S_{AP1} S_{AP2}}$ (blue) and their phase difference $\phi_{P1} = \phi_{AP1} - \phi_{AP2}$ without the linear term (yellow). $y$ Axis scale and ticks for spectral intensity match the $y$ axis of next plot. **d** Fourier-limited (blue) and phase-affected (red) AP temporal structure based on $S_{P1}$ and $\phi_{P1}$. The delay between the pulses is removed for better comparison

can be applied and the Fourier transformation (FT) of the spectrum above this energy contains information in peak P1 in Fig. 3b about the spectral phase difference of the two APs and the product of their spectra (detailed explanation of SI treatment is given in Supplementary Fig. 2 and its test with PIC simulation results in Fig. 3). Figure 3a–c represent such a range of a measured spectrum and the corresponding spectral interferometric evaluation. The product of the spectra of the two individual APs ($S_{AP1}$ and $S_{AP2}$) $S_{P1} = \sqrt{S_{AP1} S_{AP2}}$ and their spectral phase difference without the linear term are plotted in Fig. 3c. As the two dominant APs are very similar, we assume that they have the same spectrum $S_{P1}$ directly obtained from SI. Additionally assuming that the spectral phase of one of the AP (for example, first) is flat, which is in agreement with our simulations where all APs have flat spectral phase in a broad parameter range (see Supplementary Fig. 4)[49], the phase of the other one (second) becomes equal to their difference as $\phi_{P1} = \Delta\phi_{12} = \phi_{AP1} - \phi_{AP2}$, which is also obtained by the inverse FT of the peak P1. This enables to estimate the effect of the spectral phase on the AP temporal structure.

Figure 3d demonstrates that, by adding phase $\phi_{P1}$ to Fourier-limited AP with spectrum $S_{P1}$, the temporal structure remains practically unchanged, that is, our second assumption is not necessary as the two APs have the same second- and higher-order spectral phase. This was also observed in all investigated experimental shots having two APs as well as in our simulations.

It is important to note that the delay between two APs contained in the group delay (GD) of $\phi_{P1}$, which is depicted by the position of P1, is 2.7 fs, while $T_L = 2.55$ fs. Therefore, the second AP propagated some additional path and it took additional 150 as to reach the detector. Simulations predict[31,37] that plasma is pushed by the intense laser field and different APs are generated at different spatial position. Using this 150 as

propagation delay in the conventional optical path difference equations for the reflected light rays (see Methods), a spatial difference of 40 nm in normal direction is estimated.

Spectral interferometry method was thus successfully applied for the analysis of XUV spectra of two APs generated by a few-cycle laser, revealing spatial and temporal information about the process. In particular, the second- and higher-order spectral phase difference between the pulses was found to be zero. Extrapolation of this method to interference of several APs allows comparing their individual intensities and GDs, which results in a partial reconstruction of the temporal structure of the attotrain.

**Plasma surface denting**. In the $\phi_{CEP} \approx \pi$ case (and $L/\lambda_L \approx 0.4$), three APs start contributing to the XUV spectrum in the same photon energy range, which is reflected in the corresponding FT. Figure 4a illustrates such an experimental spectrum. Instead of a single peak, the corresponding FT contains three peaks originating from the non-equidistant attotrain. Each of them represents interference between different pairs of APs (Fig. 4b). As the number of peaks is the same as the number of APs, the spectrum of all three individual APs can be retrieved without any assumption as shown in Fig. 4c (see Supplementary Fig. 2 and 3). Positions of the peaks correspond to arrival time differences at the detector of interfering APs: $t_{P3} = t_{P1} + t_{P2}$. Comparison of the corresponding phase differences without the linear term (Fig. 4d) confirms this interpretation: $\Delta\phi_{13} \equiv \phi_{P3} = \phi_{P1} + \phi_{P2} \equiv \Delta\phi_{12} + \Delta\phi_{23}$. Assumption of a flat spectral phase of any AP allows representing the other two phases through phase differences $\Delta\phi_{12}$ and $\Delta\phi_{23}$. Applying such a phase to the measured Fourier-limited individual APs, their duration again does not show any AP elongation. These observations are consistent with the numerical predictions that all generated APs in the measurement were (atto) chirp-free. Thus FT provides the temporal structure of attotrain (see Fig. 6a) with the previous assumption. The ratios between the AP intensities are, however, not influenced by this assumption.

From the arrival time differences of APs at the detector, the relative spatial and temporal coordinates of the AP generation are derived, as shown in Fig. 4e. The time direction ambiguity is eliminated by comparison to simulations, where the GD difference of first and second pulse is always much larger. This curve is explained by so called plasma denting—movement of the plasma surface caused by the pushing intense laser field. Figure 4f shows the simulated electron density in the laboratory system normalized to the critical density $n/n_c$ during the interaction with the laser pulse.

The averaged shape of the plasma surface within one optical cycle is very close to the parabola formed by the birth place coordinates of APs obtained from the interferometric evaluation of the simulated spectrum, which is valid for a broad parameter range as shown in Supplementary Fig. 5. Therefore, the relative coordinates of the generation points of APs reconstruct the motion of the plasma surface. Figure 4f demonstrates such a parabolic interpolation with similar parameters as in Fig. 4e.

It is expected that softer plasma (longer $L$) experiences a stronger denting, which leads to a larger interval between peaks P1 and P2. However, for some particular cases they partially overlap or completely merge into a single peak. This is possible when the plasma is very dense and the corresponding denting is negligible. In these cases, the GD differences are derived using only this merged peak together with P3 but without the possibility to check higher-order spectral phase differences. Comparison of such reconstructions for different CEP values and plasma scale lengths is shown in Fig. 5a. The represented CEP interval corresponds to a P3 peak amplitude above noise level in the FT of

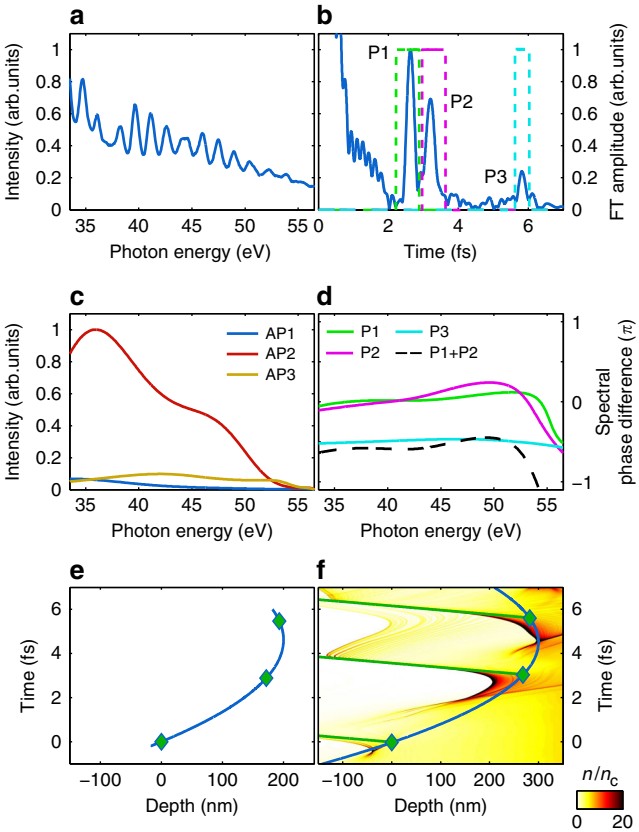

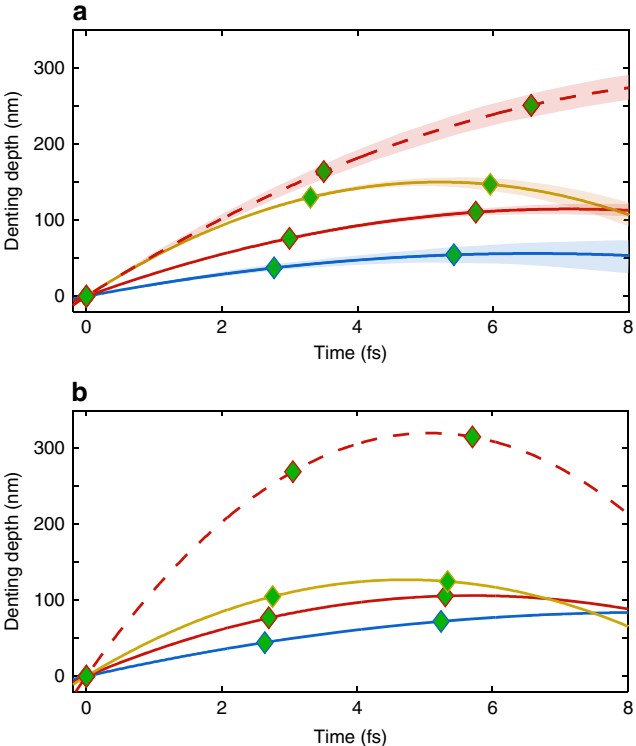

**Fig. 4** Spectral interferometry of three APs. **a** Measured XUV spectrum with $\phi_{CEP} \approx \pi$ and $L/\lambda_L \approx 0.5$ within selected spectral range. **b** Corresponding FT using the 33–57 eV range. Peaks P1, P2, and P3 represent interference of different pairs of three involved APs: AP2 & AP3, AP1 & AP2, and AP1 & AP3, respectively. Dashed lines show the applied temporal gate functions for inverse FT. **c** Reconstructed individual spectra of all three APs within the considered spectral range. **d** Corresponding spectral phase differences without the linear term. Black dashed line illustrates the sum $\phi_{P1} + \phi_{P2}$, which well matches $\phi_{P3}$. A $-\pi/2$ constant is added to $\phi_{P3}$ and the phase sum line for better visibility. **e** Reconstructed plasma surface dynamics, based on the calculated relative coordinates of three APs (see Methods). **f** Result of 1D LPIC simulation for the electron density in the plasma driven by the laser field with pulse parameters similar to the experiment ($\phi_{CEP} = \pi$ and $L/\lambda_L = 0.4$). The green diamonds mark the points where XUV pulses are generated and are obtained from the interferometric evaluation of the simulated spectrum. The color bar indicates normalized electron density $n/n_c$. The blue line is a parabolic interpolation based on these three generation points that coincides well with the averaged plasma surface denting. Green lines show propagation directions of the generated XUV pulses

**Fig. 5** Waveform-dependent relativistic plasma surface denting. **a** Parabolic interpolation for the plasma surface motion evaluated from the measured XUV spectra. Red, blue, and yellow solid lines correspond to the plasma scale length $L/\lambda_L \approx 0.13$ and respective $\phi_{CEP}$ values $7\pi/6$, $\pi/2$, and $5\pi/3$—center and the edges of the CEP range leading to a P3 peak in FT above the noise level. Red dashed line shows plasma surface denting for significantly softer plasma $L/\lambda_L \approx 0.5$ and $\phi_{CEP} \approx 7\pi/6$. Lines are fits based on the mean value of the generation points (with in $\phi_{CEP} \pm \Delta\phi_{RMS}$), shaded areas correspond to standard deviation. **b** Similar reconstruction based on the simulated spectra. CEP values are the same as above and scale lengths are $L/\lambda_L = 0.13$ for solid lines and $L/\lambda_L = 0.4$ for dashed line. For all the cases, first AP spatial and temporal coordinates are used as reference

the measured spectra, that is, there are three considerable APs present in the selected spectral range. Figure 5a shows that the increase of prepulse delay, that is, longer plasma scale length, leads to a stronger denting. It also illustrates that the reconstructed parabolas significantly depend on the CEP of the driving pulse. Variations in the depth of denting caused by CEP change are smaller than those by significant plasma scale length increase.

Nevertheless, the effect unambiguously demonstrates that, for a few-cycle driving laser pulse, even the curvature of the reconstructed parabolas without relying on their origin (overlapping coordinate) are field dependent. Therefore, it is expected that the plasma surface dynamics is also CEP dependent as

opposed to the case of long driving pulses that lead to only intensity-dependent plasma surface motion[31]. Figure 5b shows similar comparison of plasma surface denting for the simulated spectra. Both effects—significant dependence on CEP and plasma scale length—are in qualitative agreement with the experiment. This agreement together with the visual comparison between Fig. 2b and c were the criteria to choose the scale length in simulation. The differences between experiment and 1D simulations might originate from multi-dimensional effects as in the $\lambda^3$ regime[58]. We conclude that these experimental observations and simulations support the field-dependent plasma denting phenomena.

**Isolated AP**. The spectrum in Fig. 4a corresponds to a dominant central AP, which is at least five times stronger than the side APs, as visible in the calculated temporal structure in Fig. 6a. This relation is also predictable from the FT; according to the equations (Supplementary Fig. 2) stronger P1 and P2 peaks relative to P3 encode a more dominant central AP. At certain CEP values, which depend on the plasma scale length and the laser intensity, P3 vanishes below the noise level, while peaks P1 and P2 are still well observable.

It indicates that the two side APs are much weaker than the central one. Figure 6b shows such an experimental spectrum (blue line) and its FT is plotted in Fig. 6c. Here only two peaks

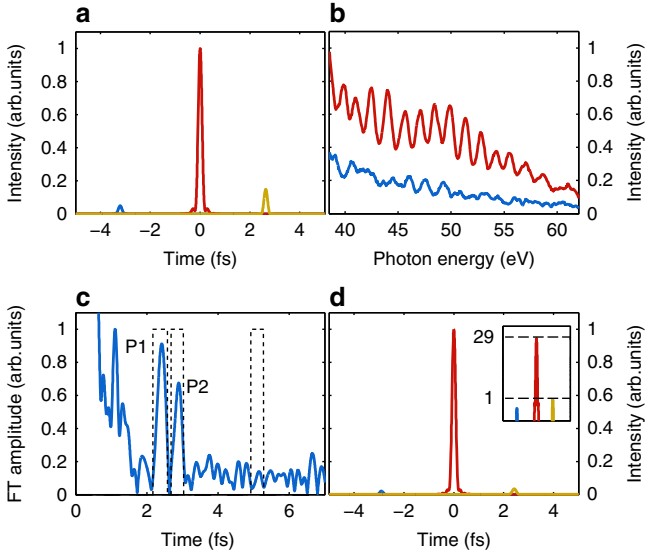

**Fig. 6** Highly isolated AP. **a** Reconstructed temporal structure of the attotrain based on the individual spectra and corresponding phase differences from Fig. 4c, d and assuming a flat phase for one of the pulses as observed in our simulations. **b** Measured spectrum (blue) with $\phi_{CEP} \approx 11\pi/6$ and $L/\lambda_L \approx 0.5$ in comparison with highly modulated spectrum (red) appearing as a result of two equivalently strong APs ($\phi_{CEP} \approx \pi$). **c** Corresponding FT of the measured XUV spectrum attributed to highly isolated AP. Peaks P1 and P2 represent interference of this dominant AP with weak side ones. P3 is below the noise level due to very low spectral intensity of the side APs. **d** Reconstructed temporal structure demonstrating minimum isolation degree of the central AP of 30. Inset: AP intensities in logarithmic scale with different normalization for better comparison

responsible for interference of the side APs with central one are visible. The position of P1 and P2 predicts, however, the position of P3. Therefore, the analysis is repeated without clear P3 peak by using noise signal at the corresponding FT position instead of real P3 amplitude. In this case, reconstruction of the individual spectra of APs is not accurate, but it only leads to under-estimation of the real isolation degree. Corresponding temporal structure with actual delays given by the spectral phase differences is presented in Fig. 6d. It demonstrates a minimum (intensity) isolation degree of 30 of the central AP, which is definitely considered as an isolated AP.

These relativistic HHG results clearly support isolated APs from solid-density plasma surfaces.

## Discussion

We have presented experimental results showing CEP-dependent high-order harmonics from relativistic PMs supporting a well isolated AP. A unique interpretation of the harmonic spectra as an SI trace of a few APs and the corresponding evaluation delivers formerly unattainable information about the spectrum and spectral phase difference between the pulses, including their temporal separation. This SI analysis reveals that the APs in the attotrain have the same second- and higher-order spectral phase, that is, all of them are compressed or chirped in the same way. Based on simulations, it is expected that the pulses are unchirped (no atto chirp). Furthermore, the analysis provides the dynamics of the denting of the solid density reflecting plasma surface without any spatial measurement. Based on this observation, the plasma motion is expected to be dependent on the field of the driver laser.

Our results open the way toward next-generation XUV and X-ray sources based on relativistic PMs. These sources provide unprecedented photon numbers by utilizing multi-TW to peta-watt laser powers, photon energies in the multi-keV range, and corresponding coherent extreme X-ray continuum supporting few-attosecond or even zeptosecond pulses. These extraordinary properties support pioneering applications, such as nonlinear attosecond physics, attosecond pump–attosecond probe spectro-scopy of atoms and molecules in the XUV and X-ray spectral region, attosecond X-ray diffraction, and time-resolved nuclear photonics.

## Methods

**Experimental set-up.** The laser energy of LWS-20 was reduced by losses on the PM and its optics ($T = 70\%$) and the vacuum beamline ($T = 76\%$). In the experimental vacuum chamber, the laser beam was focused with an $f/1.2$ OAP mirror to a spot size of $d_{FWHM} = 1.3$ μm, which was carefully optimized with feedback loops utilizing the adaptive mirror and wavefront sensor in the laser. The large fused silica target was translated after every shot to provide a fresh surface. The long-term reliable synchronization of the translation stages limited the applied repetition rate to 1 Hz, although the driving laser can operate at 10 Hz.

To tag the CEP, a small portion of the beam was reflected by a pellicle beam-splitter and directed into the phasemeter. Before all experimental runs with tagging to determine the relative CEP on-target of each laser pulse, preliminary shots were accumulated to obtain statistics about the CEP measurement error, which was typically $\Delta\phi_{RMS} = 210$ mrad. The absolute CEP on the target was obtained from the relative values by comparing the Fourier transform of the experimental CEP-sorted spectra with that from PIC simulations so that the third AP indicated by the 5–6 fs peak appears in the same CEP range.

**Scale length.** The characteristic plasma scale length $L$ is used to describe plasma expansion and defined as $n(x) = n_0 \exp(-x/L)$.

**Prepulse generation.** A small quarter-inch mirror was installed on a motorized stage in front of the last mirror before focusing. It reflected a small portion of the main beam in the same direction as the main beam after the last mirror with shorter optical path, that is, arriving before the main pulse by a certain time called prepulse delay. The prepulse mirror was inserted with its reflecting surface back-ward:[34] the beam was propagating through the thin glass layer before and after being reflected. This provided a delay that is needed for reaching zero-delay while the prepulse mirror is in front of the main mirror plane. The mirror position corresponding to zero-delay was calibrated with the interference between the main and the prepulse observed at focus. The prepulse delay was adjustable in the range from −1.5 to 12 ps, where negative values indicate a postpulse instead of a prepulse and were used as "no prepulse" case during harmonics generation. The intensity of the generated prepulse at focus was estimated to be $10^{-4}$–$10^{-5}$ of the main pulse according to pulse elongation in the glass and CCD signal at focus, which indicated a 6 times larger spot size and 100 times lower energy as the main pulse. Supple-mentary Fig. 6 demonstrates dependence of the measured XUV signal intensity on the prepulse delay. Zero-level signal without prepulse (at −0.33 ps) hints at a too steep plasma for the ROM generation, that is, satisfactory good high-dynamic-range contrast of the main pulse. The obvious maximum around 3 ps and following degradation (up to five times) proves the usability of this method for controlling of the plasma scale length. The scale length for a certain prepulse delay was obtained by hydrodynamic simulations with the MEDUSA code[56].

**XUV spectra contrast enhancement.** Spectral amplitude of the measured XUV signal falls very rapidly with increasing photon energy. In order to represent harmonics within a broad spectral range in one figure and observe their photon energies easier, the following steps were consequently applied to each individual spectrum: (1) bandpass filtering (between 1 and 12 fs); (2) magnification of higher photon energy part by dividing the spectrum by a fourth-order polynomial fit; and (3) normalization to the maximum oscillation amplitude. It was carefully checked that none of these modifications changes the photon energy of harmonics.

**PIC simulation.** The 1D PIC simulations are performed using the code LPIC++[7]. The incident laser has an electric field waveform $E_y^{inc} = a_L \exp\{-2\ln 2[(t - t_p)/\tau_L]^2\} \cos[2\pi(t - t_p)/T_L + \phi_{CEP}]$, with $a_L$ the nor-malized vector potential, $\tau_L$ the intensity FWHM pulse duration normalized to the laser period $T_L = \lambda_L/c$, $\phi_{CEP}$ laser CEP, and the field direction is defined for a cosine pulse that the most intense half-cycle points outwards the target. Throughout this paper, $a_L = 6$ is assumed that corresponds to a laser intensity of $8.4 \times 10^{19}$ W cm$^{-2}$ for a laser central wavelength $\lambda_L$ of 765 nm. The density profile of the interacting plasma has an exponential interface layer in front of a constant density slab layer. The density of the interface layer rises from $0.1n_c$ up to a maximum of $400n_c$, that is, the density of glass targets when fully ionized, with a scale length of $L$, where

$n_c$ is the critical electron density at the laser wavelength. The slab layer has a thickness of $2\lambda_L$. The p-polarized laser pulse is incident onto plasma at an angle $\alpha_{inc} = 55°$. In LPIC++, this oblique incidence geometry is transformed into a 1D case using the Bourdier technique[59].

**Plasma denting estimation.** For two light rays being reflected in vacuum from two parallel planes with $\Delta x$ normal coordinate difference with the given incidence angle (see Supplementary Fig. 7), the optical path difference ($\Delta d_{opt}$) and the corresponding delay ($\Delta t_{arr}$) appears to be $\Delta d_{opt} = 2\Delta x \cos(\alpha_{inc}) = c\Delta t_{arr}$, which together with $T_L$ delay leads to the arrival time difference at detector between two consecutive APs $\Delta\tau_{arr} = T_L + \Delta t_{arr} = T_L + \Delta d_{opt}/c$. Assuming AP generation happens simultaneously with the reflectance of laser pulses, that is the APs are generated at the same phase in the different optical cycles of the driving laser, and their further propagation is coincident, we can attribute a position of corresponding peak in FT to this time difference. Thus spatial and temporal coordinates of AP generation (relative to each other) can be expressed as

$$\Delta x = (\Delta\tau_{arr} - T_L)c/[2\cos(\alpha_{inc})]$$
$$\Delta t_{gen} = T_L + \Delta x \cos(\alpha_{inc})/c.$$

For an attotrain of 3 pulses using the first point as an origin ($t = 0$, $x = 0$), the other two points with their respective relative coordinates $\Delta x$ and $\Delta t_{gen}$ define a parabola that describes cycle-average plasma surface denting in $y = 0$ plane of the laboratory system. The assumptions are supported in the simulations by the good agreement between the plasma surface motion and the evaluated parabolas (see Fig. 4f and Supplementary Fig. 5).

## Data availability

The data that support the findings of this study are available from the corresponding author upon reasonable request.

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

## Acknowledgements

We thank Ferenc Krausz and George D. Tsakiris for their encouragement and motivation. This work was supported by DFG Project Transregio TR18 and The Munich Centre for Advanced Photonics (MAP), as well as within the framework of the EUROfusion Consortium and has received funding from the Euratom research and training programme 2014–2018 under grant agreement 633053. L.V. acknowledges the support by a grant from the Swedish Research Council (2016-05409). G.M. was supported by China Postdoctoral Science Foundation (Grant No. 2017M622619). Part of the work of M.A. and I.B.F. was supported by Laserlab Europe, project MPQ002119.

## Author contributions

A.B. and L.V. conceived the experiments; D.K. and A.B. operated the plasma mirror; and D.K., A.B., G.M., W.D. and M.A. conducted the experiments. B.B. contributed to the CEP tagging measurement and evaluation. G.M. performed PIC simulations and D.K., G.M., A.B., I.B.F. and L.V. interpreted the experimental data. L.V. supervised the project. D.K., G.M., M.A., I.B.F. and L.V. wrote the manuscript and all the other coauthors contributed to finalizing the paper.

## Additional information

**Competing interests:** The authors declare no competing interests.

