## [Peer Review File · Nature Communications]

Reviewers' comments:

Reviewer #1 (Remarks to the Author):

Synopsis

The authors of this article report on the spectral analysis of high-order harmonic (HH) fields generated with relativistically oscillating plasma mirrors (ROMs) on a solid target. By tagging the carrier envelope phase (CEP) of the two-cycle fundamental laser pulse in each laser shot for high-order harmonic generation (HHG), they reveal that the temporal structure of the emitted relativistic high-order harmonic field should depend on the CEP of the fundamental laser pulse. The authors also attribute the unequal temporal separations between the multiple attosecond pulses formed by the HH field to the dynamical change of the plasma surface (denting) by comparing experimental data and simulation results.

In the introduction, the authors refer to the several kinds of fundamental concept of HHG from a laser plasma formed on the surface of a solid target (coherent wake emission (CWE), ROM, coherent synchrotron emission (CSE), and relativistic electronic spring (RES)), and then state that they report the first demonstration of the CEP dependence of the HHG from the ROM, in contrast to the CEP dependence of the HHG from the CWE demonstrated by another research group.

In the next section (Results), the experimental details and analysis are described. The fundamental laser pulse is delivered from an optical parametric chirped pulse amplification (OPCPA) system named "Light Wave Synthesizer 20 (LWS-20)", whose temporal background contrast against the pulse peak is reduced by a factor of 300 with a plasma mirror set in front of an optical unit to control the delay of the prepulse forming the preplasma on the target. The prepulse and the main pulse are focused with an off-axis parabolic (OAP) mirror and the resultant intensity of the main pulse on the target is estimated to be 1.3×10^{20} W/cm², which is sufficiently high to generate the HH from the ROMs because the normalized vector potential calculated from this intensity and the laser wavelength (796 nm) is much larger than unity.

The authors record each HH spectrum at each laser shot with a single-shot stereo above-threshold ionization spectrum of electrons to discriminate the CEP of each laser shot. The HH spectra tend to monotonically decrease with the increase of the photon energy and they are consistent with the power scaling law for the photon energy theoretically obtained by assuming the HH to be generated from the ROM. Therefore, the authors conclude that origin of the measured HH should be the ROM. The authors also find that the phase of the sinusoidal modulation in the HH spectra with a period of the photon energy of the fundamental laser field alters in accordance with the CEP, resulting in the opposite phase of the sinusoidal modulation at the π -CEP compared with that at the 0-CEP. The authors claim that this is the first experimental evidence for a relativistic waveform-dependent interaction because the measured phase shift of the modulation is in good agreement with the simulated result based on the theoretical model of the ROM.

In the following subsection, the modulation in the HH spectrum is analyzed from the viewpoint of the spectral interference of multiple attosecond pulses synthesized with the HH field. The authors first apply the inverse Fourier transform (IFT) to the HH spectrum with a CEP of $\pi/3$ and a plasma length of $0.25\lambda_L$, and demonstrate that only a single peak at 2.7 fs appears in the resultant time-domain spectrum. This is owing to the fact that there are two attosecond pulses with a temporal separation of 2.7 fs. The authors obtain the relative spectral phase difference between the two attosecond pulses and the spectrum by performing the Fourier transform (FT) of the gated temporal spectrum around the peak. As a result, the dispersion of one attosecond pulse is almost identical to that of the other. The discrepancy of the time separation of 2.7 fs from the optical period of the fundamental laser field, 2.55 fs, is regarded as a proof of a 0.15 fs propagation time delay caused by the change of the plasma position pushed by the intense laser field.

Imposing another CEP and plasma length, the IFT of the HH spectrum exhibits three distinct peaks in the time domain spectrum. Each peak indicates each time separation between two of the three attosecond pulses contained in the HH field, which should be unequal to each other. The authors retrieve each spectrum of the three attosecond pulses and each relative phase difference between the two from the analysis of the FTs of the gated time domain spectra around the three peaks. The relative time delays of the second and third attosecond pulses to the first attosecond pulse are converted to the equivalent relative positions at the times when the attosecond pulses are emitted. The authors attribute the time delays to the denting of the plasma surface emitting attosecond pulses because the equivalent relative positions of the attosecond pulse emitters estimated from the experimental data agree with the positions obtained from a simulation demonstrating the denting of the plasma surface emitting attosecond pulses. Based on this assumption, the authors find that the denting depth changes by scanning the CEP within a range where the three distinct peaks can be observed in the time domain spectrum. The authors claim that this is evidence of field-dependent plasma denting.

In the final subsection (Results), the authors show that the third peak in the time domain spectrum disappears with a CEP of $11\pi/6$ and a plasma length of $0.5\lambda_L$. This result originates from the significant reduction of the intensities of both the first and third attosecond pulses, and hence, the authors claim that an isolated attosecond pulse is generated for the first time from solid-density plasma surfaces.

In the final section (Discussion), the authors summarize the conclusions of the subsections in the previous section. They also state the prospects for applications of the HHs from the ROM to nonlinear attosecond physics and time-resolved spectroscopies in the XUV and X-ray spectral regions.

Comments

I acknowledge that the experimental results described in this article are very important for the scientific communities studying the HHG from plasmas interacting with ultrashort laser pulses at relativistic intensities. In particular, the phase shift of the HH modulation with the CEP alteration depicted in Figs. 2(a) and (b) should be recognized as clear evidence for the waveform-dependent interaction of ultrashort laser pulses in a relativistic regime. In addition, the Fourier analysis of the spectral interference fringes appearing in the generated HH spectra provides the relevant information of the attosecond pulse train and the dynamical evolution of the plasma surface.

Therefore, I recommend this article to be published in *Nature Communications* if the authors accept the revision of the manuscript to address the following technical issues.

1. On page 1, left column, line 8, “*Its main advantage over gas HHG is . . . higher energy and shorter wavelength.*”

I acknowledge that the HHG from plasma mirrors is advantageous owing to the fact that the pulse energies of HH pulses are estimated to be much more than $1 \mu\text{J}$ in the experiments reported in refs. [20], [21] and [22]. The authors should address the pulse energy or the conversion efficiency of the HH pulses obtained in their experiment somewhere in the manuscript if they would like to convince readers of the authors’ claim, because the wavelength range of the measured HH from ROMs is similar to that of conventional gas HHs.

2. On page 2, right column, line 21, “. . . in an angle of incidence $\alpha_{inc} = 55^\circ$ and p-polarization.”

Why did the authors configure the incident angle to be 55° in the experiment, although they fixed the incident angle to be 45° in the numerical simulation reported in ref. [44]?

- 3. On page 3, left column, line 32, “The harmonic spectra were fitted . . . the theoretically expected ROM scaling of $I_\omega : \omega^{-8/3}$.”**

The decay of spectral intensity of the HHs in accordance with the power law of the photon energy, $\omega^{-8/3}$, is significant evidence of the ROM HHG. Therefore, the authors should exhibit a fitting curve and the fitted HH spectrum so that readers of this article can acknowledge that the measured HH spectrum is consistent with that obtained from a theory based on the ROM model.

- 4. On page 3, left column, line 45, “Sorting the measured spectra according to the corresponding laser pulse ϕ_{CEP} reveals continuous shift of the photon energy of harmonics.”**

The definition of ϕ_{CEP} is missing. Even though the authors adopt an equation for E_y^{inc} in ref. [44], the direction of the electric field of the incident laser pulse is not unambiguously defined because the direction of the y axis is not identified. The authors should clarify which is the correct 0-CEP pulse in the following two figures.

[Redacted]

- 5. Figure 2(a) and Figure 2(b)**

It would be beneficial for readers if the authors provide more detailed information of data acquisition and analysis, such as the number of shots for accumulating HH spectra and the width of bins used to discriminate the CEP.

- 6. On page 4, left column, line 28, “. . . the Fourier transformation (FT) of . . . information in peak P1 in Fig. 3b . . .”**

Two labels of “P1” are placed outside Fig. 3(b). The authors should put a correct “P1” label in Fig. 3(b).

- 7. On page 4, left column, line 41, “Additionally assuming the spectral phase of the first AP is flat, which is in agreement with our simulations⁴⁴ . . .”**

I could not find spectral phases of two almost equivalent attosecond pulses in the figures shown in ref. [44]. The authors should correctly refer to references or exhibit the spectral phases of the two almost equivalent attosecond pulses obtained from the simulation in this article.

8. On page 5, left column, line 13, “. . . , three APs start contributing to the XUV spectrum in the same photon energy range, . . . ”

The photon energy range of the horizontal axis in Fig. 3(a) is not the same as that in Fig. 4(a). The authors should explicitly indicate the photon energy range to be analyzed with the FT in both figures.

9. Figure 4(c) and 4(d)

There is no specification for the traces with red, yellow, and blue colors in these figures. The authors should identify each trace with each attosecond pulse.

10. On page 5, right column, line 1, “The averaged shape of the plasma surface, . . . ”

What kind of average did the authors apply to the plasma surface?

Reviewer #2 (Remarks to the Author):

The manuscript “Spectral interferometry with waveform-dependent relativistic high-order harmonics” by D. Kormin et al is an experimental study of the interaction of a high-intensity laser field with an ionized solid surface. Studying this process is strongly motivated by the opportunity to produce the intense, tailored, isolated pulses with uniquely short duration (in the attosecond range) that can be used for fundamental studies of electron dynamics in atoms and molecules. The use of a two-cycle laser pulse with varied carrier-envelope phase and peak intensity high enough to drive relativistic motion in electrons makes it possible for the authors to excite tailored relativistic plasma dynamics. According to earlier theoretical studies of this scenario, in this relativistic regime the plasma electrons tend to form a clearly distinguished plasma-vacuum interface, the dynamics of which is responsible for the production of bursts of XUV radiation. The authors use spectral interferometry to retrieve information about the relative intensity and duration of, and the temporal delays between, the two or three most intense bursts of radiation. In this way, for the first time, the formation of the plasma-vacuum interface and its dynamics in the relativistic regime were observed and temporally resolved, albeit indirectly. The authors relate their results to phenomenological descriptions and ab initio numerical simulations to a certain degree of success, and further demonstrate the controllable generation of an attosecond pulse with a remarkable degree of isolation. Although the spectral interferometry procedure used in this work was developed to a large extent in earlier work (Ref. 46), the results here are essential for understanding and controlling the laser-plasma interaction at highly relativistic intensities and will influence thinking in this field. In this respect, I find that the results are sufficiently impactful to meet the criteria of Nature Communications. Nevertheless, some of the interpretations and claims are not convincing enough that I can recommend the current version for publication. These questions and concerns are:

1. The authors write that “the harmonic spectra were fitted for many shots and found to scale with the harmonics frequency in a power close to the theoretically expected ROM scaling...”, i.e. a power law with exponent of $-8/3$. On this basis, the authors conclude that “in the observed spectral range of 16-100 eV XUV emission was dominantly generated via the ROM mechanism.” This conclusion is an unnecessarily oversimplified treatment of a problem that still lacks a clear theoretical understanding/description. The ROM mechanism is not derived ab initio, but from a phenomenological treatment of the plasma-vacuum interface. The applicability of this phenomenological treatment to the considered here case, where the density rises gradually, is questionable, while there could be other explanations for the harmonic generation that lead to similar spectral properties. Given the theoretical considerations given in, for example, Pirozhkov et al. Phys. Plasmas 13, 013107 (2006), Debayle et al.

Phys. Plasmas 20, 053107 (2013), Boyd & Ondarza-Rovira, Phys. Rev. Lett. 101, 125004 (2008) and Phys. Lett. A 380, 1368 (2016), it would be useful to provide the actual range of exponents (of the power law) that fit the data, or better still, the measured spectra themselves in the relevant frequency range.

2. The authors interpret the additional delays they observe in the attosecond pulse (AP) generation as a consequence of plasma denting. This interpretation implies that the phase of AP generation, relative to the phase of the incident wave, is the same for each of the radiation cycles that contributes. Even under the assumption of relativistic similarity, this phase should depend on the ratio of the local plasma density to the wave amplitude (see, for example, fig. 5(a,d) of Ref. 13). In general, this ratio can be different for different cycles, as the amplitude changes quickly and the incident radiation will interact with plasma of varying density, due to the very denting the authors describe (although the variation of amplitude and effective density might compensate each other to some degree). This might be one of the reasons underlying the carrier envelope phase (CEP) dependence that is observed, as different CEP implies different effective amplitudes of the incident wave for such a short laser pulse. Nevertheless, even the inclusion of this factor might be insufficient, as for such short laser pulses different parts of each laser cycle have different amplitudes, and this materially affects the plasma dynamics. The CEP dependence indicates that this is likely the case. Although denting seems to be a reasonable first-order explanation for the scenario at hand, the authors should provide either estimates or motivation for their neglect of the factors I have outlined, or soften their claims of interpretation. I would also encourage them to present the actual dependence of the AP spacing retrieved in order that future theoretical works might be benchmarked against this data.

3. As the laser pulse is short and the ions do not move significantly during the interaction, the plasma denting should be more or less symmetric, i.e. both forward and backwards motion of the plasma surface should be included. However, all the dependencies retrieved show only consecutive forward displacement of the points of XUV pulse generation. This is rather puzzling – can the authors explain it?

Minor comments:

1. The title does not explicitly indicate that the results are related to high harmonic generation from a plasma surface. The “spectral interferometry” that is stressed instead does not seem to be the paper’s strongest quality to me, in terms of novelty and interest.

2. I am not entirely sure that I completely understand the rationale behind using the term “waveform-dependent”. If the authors have a reason to differentiate “waveform-dependent” from “CEP-dependent”, this should be clarified, ideally in the abstract.

3. It is worth mentioning in the main text (not in supplementary) that the plasma scale-length was linked to the prepulse setting by MEDUSA simulations, and a reference should be provided.

4. The authors state that “each recorded XUV spectrum was tagged with the CEP of the corresponding laser pulse, measured with a single-shot stereo above-threshold-ionization (ATI) phasemeter (CEP-meter).” However, they also state that “the absolute CEP in experiments was determined by comparing simulations and experiments.” Thus it is not clear how they traced the CEP.

5. When specifying the isolation degree, it is worth mentioning that it is for intensity, not for amplitude.

6. On page 5 the authors write “Fig. 4f demonstrates such interpolation with similar parameters as in Fig. 4b.” Do the authors mean Fig. 4e in the latter reference?

7. On page 7 the authors write “in this case reconstruction of the individual spectra of APs is not accurate, but it only leads to underestimation of the real isolation degree.” Why does this lead only to

underestimation? It is not clear to me.

Reviewer #3 (Remarks to the Author):

This manuscript reports on an experimental and theoretical investigation on attosecond train pulses generated from plasma mirrors in the relativistic regime driven by a two-cycle laser pulse. Due to the motion of the plasma surface during the interaction, the attosecond trains are not equally separated in time, which can allow to apply spectral interferometry to disentangle the cross-correlation of pair of pulses from the HHG spectrum, and get more insight on the generating process than what is possible in HHG from a gas medium. The paper is well-written and suitable for a broad audience. In particular, the “plasma surface denting” section clearly shows how the denting depends on the CEP of the pulse. In this case, the interferometry method can be used to retrieve interesting results. However, some major issues should be dealt with prior to acceptance. In particular, I have some concerns about some statements and the data analysis shown in the “spectral interference” section. Hereafter a list of the main critics is reported.

major issues:

1. through the paper, different plasma scale lengths are used in the simulation than the experimental ones obtained from the MEDUSA code given a certain pre-pulse delay. Is there some physical reason why the authors chose a different parameter or it is just to fit the experiment? This should be clarified. In particular, for the $L/\lambda = 0.13$ case, the simulations in figure 2 have been done at $L/\lambda = 0.1$ while in figure 5 they have been done at $L/\lambda = 0.15$. While I can understand that there could be a disagreement between the scale length derived with the MEDUSA code and the one used in the simulations, I cannot understand why choosing a different theoretical number (0.1 and 0.15) for the two cases. For the same reason, I would not say that “Both effects.. are in perfect qualitative agreement with the experiment” as stated at the end of the “Plasma surface denting” section. It would be useful to show how much the theoretical results change when you use the L/λ parameter retrieved from the experimental pre-pulse delay and compare it with the one that fits the data, in order to let the reader understand what is the order of magnitude of the disagreement if there is any.

2a. In the “Spectral Interference” section the authors assume that the spectral phase of the first AP is flat and so they claim they can measure the phase of the second pulse. They refer to previous simulations from Ma G et al. [Phys. Plasmas 22, 24879 (2013)] where no clear information on the second and third order of the phase of the generated pulses seems to be given. The flat phase can be argued from FIG.1 of that paper, where the attosecond burst seems to have the same delay for every photon energy, but it is difficult to estimate second and third order terms from this graph (and in any case FIG.1 refers to a different CEP value). It’s not clear why we could not assume that the phase of the second pulse is flat and so retrieve the phase of the first one. Are second and third order terms greater in the second pulse? Is this supported by theoretical simulations previously done?

2b. Dropping the assumption that the phase of the first pulse is flat, the only thing that can be measured, as the authors state, is the difference of phase between the two pulses. This difference is shown in figure 3c. However, this phase is not easy to read. The linear dependence must be subtracted in order to show the differences in the second and third order terms of the two pulses. Even if the duration is not affected so much, as shown in figure 3d, some wings appear in the pulse, which suggests the presence of higher order terms that cannot be appreciated from figure 3c if the linear phase is not subtracted. A comparison of this second and third order terms with theoretical predictions would add value to this section, focusing more on what can be actually measured.

3. In the section “Plasma surface denting”, assuming that one attosecond pulse has a flat-phase, the authors can retrieve the phase of the other two pulses showing that they also have a linear phase. Again, no clear reference is given on this assumption. As a consequence, some sentences should be rephrased, like “These observations indicate that all generated APs in the measurement were chirp-free”.

minor issues:

1. please give a definition of the normalized vector potential, it could be useful for a broad audience

2. I do not understand why there are different CEPs values in the text and in the captions. For example, in the “spectral interference” section, $\Phi_{\text{CEP}} = 0$ according to the text, while it is $\pi/3$ according to the caption. Also in the “plasma surface denting section”, $\Phi_{\text{CEP}} = \pi$ in the text, while it is equal to $2\pi/3$ in the caption.

To conclude, I believe that some approximations, especially in the “spectral interference” section, must be better justified. I’m overall favorable to the publication of these results in Nature Communications after the authors have addressed the issues raised.

Reviewer #1

We thank the referee for the review and appreciating our results. We revised the manuscript including the technical issues listed by the referee. In the answers we used the reference numbers of the original (not resubmitted) manuscript.

1. On page 1, left column, line 8, "Its main advantage over gas HHG is . . . higher energy and shorter wavelength."

The authors should address the pulse energy or the conversion efficiency of the HH pulses obtained in their experiment somewhere in the manuscript if they would like to convince readers of the authors' claim, because the wavelength range of the measured HH from ROMs is similar to that of conventional gas HHs.

We agree with the referee and also wanted to measure the energy, but the energy diode was saturated and didn't allow any conclusions about the XUV energy. However, at this point in the introduction we intend to say that in general higher energy is observed (see refs 20-21, 23, 30) not in our specific case. This higher energy is generally valid even if the (strongly debated) conversion efficiency is comparable to gas HHG, however, if it is higher the difference further increases. This is mainly due to the much higher input laser energy that is not applicable in gas harmonics experiments.

2. On page 2, right column, line 21, ". . . in an angle of incidence $\alpha_{inc} = 55^\circ$ and p-polarization."

Why did the authors configure the incident angle to be 55° in the experiment, although they fixed the incident angle to be 45° in the numerical simulation reported in ref. [44]?

Large number of simulations were performed before the experiment with 45° based on ref [44]. However, experimental limitations forced us to use 55° . To investigate the difference some 55° simulations were performed originally as shown in the figure below illustrating the measured CEP map

(y-axis is CEP in π) and the simulated ones for 45° and 55° (a_{inc}) and two different scale length values ($L=0.1\lambda$ and $L=0.15\lambda$). No significant influence is observed.

However, as this relative small deviation in the angle of incidence might have a large influence on the attosecond pulses and the generation mechanism [see for example refs 12-14] we decided to perform (a large number of) new simulations with 55° .

All relevant simulations in the manuscript are replaced

by these new results (only some examples in the Supplementary using 45°), which caused smaller changes in the figures. However, the main outcome is the same as before and there is no relevant influence of the slightly larger angle of incidence.

3. On page 3, left column, line 32, "The harmonic spectra were fitted . . . the theoretically expected ROM scaling of $I_w : w^{-8/3}$."

The decay of spectral intensity of the HHs in accordance with the power law of the photon energy, $w^{-8/3}$, is significant evidence of the ROM HHG. Therefore, the authors should exhibit a fitting curve and the fitted HH spectrum so that readers of this article can acknowledge that the measured HH spectrum is consistent with that obtained from a theory based on the ROM model.

We very appreciate the comment of the referee. To respond, we have made detailed statistics on the fit parameters for a large number of measured spectra. As an example, the fit results for $L/\lambda=0.13$ (prepulse, PP=0.5 mm) are shown below. The upper row displays two fits on two single shots and the lower row shows the exponent value of the power fit and the R^2 coefficient of determination (fit quality parameter). The exponent of the power law fit is -2.55 with a standard deviation of 0.21 for 71 shots (all CEPs), and with an R^2 of 0.95. The results for longer scale length are:

$L/\lambda=0.25$ case: exponent of the power law fit is -2.27 with a standard deviation of 0.17, $R^2: 0.95$

$L/\lambda=0.4$ case: exponent of the power law fit is -1.87 with a standard deviation of 0.21, $R^2: 0.87$

The fit quality is lower (lower R^2) for the longest scale length, because it is influenced more by the beating of 3 pulses due to the larger denting depth and thus the spectrum deviates more from the power law behavior. As a conclusion we have an exponent about -2.55 ± 0.21 for $L/\lambda=0.13$. Still, for longer scale length values, other models might provide a better description than the ROM model, which is an interesting research topic for future investigations. We have inserted an example fit in Figure 2 and the statistics about the exponent into the text and Supplementary.

4. On page 3, left column, line 45, "Sorting the measured spectra according to the corresponding laser pulse ϕ_{CEP} reveals continuous shift of the photon energy of harmonics."

The definition of ϕ_{CEP} is missing. Even though the authors adopt an equation for E_y^{inc} in ref. [44], the direction of the electric field of the incident laser pulse is not unambiguously defined because the direction of the y axis is not identified. The authors should clarify which is the correct 0-CEP pulse in the following two figures.

The CEP was defined by an equation a bit hidden in the PIC simulations Methods section. We cited the Methods section when the CEP is introduced in the text. Furthermore, we have modified the text in Methods to unambiguously define the electric field direction by inserting “and the field direction is defined for a cosine pulse that the most intense half-cycle points outwards the target”.

5. Figure 2(a) and Figure 2(b)

It would be beneficial for readers if the authors provide more detailed information of data acquisition and analysis, such as the number of shots for accumulating HH spectra and the width of bins used to discriminate the CEP.

We used single shot CEP tagging and sorted 71 measurements in Fig. 2(b). There were no direct discriminator in the CEP tagging, but we had a CEP measurement error of ± 210 mrad as mentioned in the Experimental setup Methods section. We smoothed experimental and simulated data with running average in Fig. 2(b, c) within this 420 mrad error range. To make it more clear, we inserted in the caption of Fig 2 “(b) CEP-sorted experimental XUV spectra of 71 shots for a prepulse delay of $\tau_{\text{pp}} = 1.67$ ps ... simulated spectra for better visualization of harmonics photon energy. A vertical running average smoothing of the data in (b) and (c) was applied within the CEP measurement error range of 2×210 mrad.”

6. On page 4, left column, line 28, “... the Fourier transformation (FT) of... information in peak P1 in Fig. 3b...”

Two labels of "P1" are placed outside Fig. 3(b). The authors should put a correct "P1" label in Fig. 3(b).

We thank the referee pointing out this mistake. It has been corrected.

7. On page 4, left column, line 41, “Additionally assuming the spectral phase of the first AP is flat, which is in agreement with our simulations⁴⁴...”

I could not find spectral phases of two almost equivalent attosecond pulses in the figures shown in ref. [44]. The authors should correctly refer to references or exhibit the spectral phases of the two almost equivalent attosecond pulses obtained from the simulation in this article.

Indeed ref [44] contains the spectrogram of the reflected pulse only for a dominant attosecond pulse. To respond, we put an emphasis on this point and prepared extra figures as our evaluation utilizes the flat spectral phase. We inserted two new figures in the supplementary: Fig. S2 illustrates this case (one dominant attosecond pulse) better by showing the spectral phase, and Fig. S3 shows the flat phase behavior for a large parameter range (also two almost equivalent pulses) that we observed for all simulated laser CEP and scale length values. We have inserted a short text claiming that the pulses are chirp free for all CEPs in the revised manuscript.

8. On page 5, left column, line 13, ". . . , three APs start contributing to the XUV spectrum in the same photon energy range, . . . "

The photon energy range of the horizontal axis in Fig. 3(a) is not the same as that in Fig. 4(a). The authors should explicitly indicate the photon energy range to be analyzed with the FT in both figures.

In case of Fig. 4(a) the same photon energy range means that all the 3 attosecond pulses have enough spectral intensity (that they are visible) in the plotted range (~33-57 eV) which is slightly different from the range given in Fig. 3a. Certainly, depending on the parameters (CEP, scale length etc.) the spectral range (especially the lower limit) must be matched to get 1, 2 or 3 attosecond pulses. We included the analyzed photon energy range in the corresponding captions.

9. Figure 4(c) and 4(d)

There is no specification for the traces with red, yellow, and blue colors in these figures. The authors should identify each trace with each attosecond pulse.

It has been corrected.

10. On page 5, right column, line 1, "The averaged shape of the plasma surface, . . . "

What kind of average did the authors apply to the plasma surface?

The electron density oscillates in and out within one optical cycle and this is an average over these oscillations similarly to Ref 26. We inserted the text "The averaged shape of the plasma surface within one optical cycle is very close to the parabola formed by the birth place coordinates of APs." to clearly describe this averaging.

A slight modification in the equation of the temporal coordinate of the evaluated parabola has been made, which now describes correctly 2D as well as boosted 1D geometry transformed back to the lab system. This did not influence the agreement between the simulation and experiment as the same formerly old and now new equation is used for both of them. The new 55 deg angle of incidence and $L/\lambda=0.13$ scale length simulations in Fig. 5 have a stronger influence, but still provide good agreement.

We hope our answers and revision of the manuscript is satisfactory and the reviewer finds our paper suitable for acceptance.

Reviewer #2

We thank the referee for the review and appreciating our results. We revised the manuscript implementing the questions and concerns of the referee. In the answers we used the reference numbers of the original (not resubmitted) manuscript.

1. The authors write that “the harmonic spectra were fitted for many shots and found to scale with the harmonics frequency in a power close to the theoretically expected ROM scaling...”, i.e. a power law with exponent of $-8/3$. On this basis, the authors conclude that “in the observed spectral range of 16-100 eV XUV emission was dominantly generated via the ROM mechanism.” This conclusion is an unnecessarily oversimplified treatment of a problem that still lacks a clear theoretical understanding/description. The ROM mechanism is not derived ab initio, but from a phenomenological treatment of the plasma-vacuum interface. The applicability of this phenomenological treatment to the considered here case, where the density rises gradually, is questionable, while there could be other explanations for the harmonic generation that lead to similar spectral properties. Given the theoretical considerations given in, for example, Pirozhkov et al. Phys. Plasmas 13, 013107 (2006), Debayle et al. Phys. Plasmas 20, 053107 (2013), Boyd & Ondarza-Rovira, Phys. Rev. Lett. 101, 125004 (2008) and Phys. Rev. Lett. A 380, 1368 (2016), it would be useful to provide the actual range of exponents (of the power law) that fit the data, or better still, the measured spectra themselves in the relevant frequency range.

Indeed, there are various theoretical considerations to describe relativistic high harmonic generation, which references fit well into our introduction. Furthermore, we made detailed statistics on the fit parameters for a large number of measured spectra. As an example the fit results for $L/\lambda=0.13$ (prepulse, PP=0.5 mm) are shown below. The upper row displays two fits on two single shots and the lower row shows the exponent value of the power fit and the R^2 coefficient of determination (fit quality parameter). The exponent of the power law fit is -2.55 with a standard deviation of 0.21 for 71 shots (all CEPs), and with an R^2 of 0.95 .

The results for longer scale length are:

$L/\lambda=0.25$ case: exponent of the power law fit is -2.27 with a standard deviation of 0.17, R^2 : 0.95

$L/\lambda=0.4$ case: exponent of the power law fit is -1.87 with a standard deviation of 0.21, R^2 : 0.87

The fit quality is lower (lower R^2) for the longest scale length, because it is influenced more by the beating of 3 pulses due to the larger denting depth and thus the spectrum deviates more from the power law behavior. As a conclusion we have an exponent about -2.55 ± 0.21 for $L/\lambda=0.13$. Still, for longer scale length values, other models might provide a better description than the ROM model, which is an interesting research topic for future investigations. We have cited other relevant theory papers and inserted an example fit in Figure 2, which already plots two typical spectra in the relevant spectral range, and added the statistics about the exponent into the text and Supplementary.

2. The authors interpret the additional delays they observe in the attosecond pulse (AP) generation as a consequence of plasma denting. This interpretation implies that the phase of AP generation, relative to the phase of the incident wave, is the same for each of the radiation cycles that contributes. Even under the assumption of relativistic similarity, this phase should depend on the ratio of the local plasma density to the wave amplitude (see, for example, fig. 5(a,d) of Ref. 13). In general, this ratio can be different for different cycles, as the amplitude changes quickly and the incident radiation will interact with plasma of varying density, due to the very denting the authors describe (although the variation of amplitude and effective density might compensate each other to some degree). This might be one of the reasons underlying the carrier envelope phase (CEP) dependence that is observed, as different CEP implies different effective amplitudes of the incident wave for such a short laser pulse. Nevertheless, even the inclusion of this factor might be insufficient, as for such short laser pulses different parts of each laser cycle have different amplitudes, and this materially affects the plasma dynamics. The CEP dependence indicates that this is likely the case. Although denting seems to be a reasonable first-order explanation for the scenario at hand, the authors should provide either estimates or motivation for their neglect of the factors I have outlined, or soften their claims of interpretation.

We also observed in our simulations that the process is very complex as the referee described. There might be other effects such as different phases of generation that influence the parabolas. However, as we will show assuming a constant phase is a good approximation for the plasma surface motion. We plot in the figure below the electron density for $L/\lambda=0.4$, $CEP=120^\circ$ and two different parabolas. Red

parabola is the interferometric evaluation of the simulated spectrum as the others in the manuscript, while the blue parabola is the fit for the first 3 points obtained from the crossings of the incident laser field at a constant phase (of $\pi/2$) and the emitted attosecond pulses marked with green lines [see also answer to question 3]. Here, the exact phase value ($\pi/2$) is unimportant as we obtain with a different constant phase the same parabola translated in space-time. The two parabolas agree very well and both parabolas agree well with the density plot, i.e., they provide the solid plasma surface motion with a suitable accuracy. We also checked for other CEP as well as other scale length values and the parabolas obtained this way always agreed well with each other and with the motion of the plasma density surface (see Supplementary Figure S4). Consequently a constant generation phase is an acceptable assumption.

The question “why do we obtain this good agreement, although, the amplitude and the effective density varies from cycle to cycle that

should have an influence based on ref. 13?” could be answered the following way. (a) our S similarity parameter is >10 (~ 67) using the unperturbed peak density, which is a parameter range, where the RES model of ref. 13 is not valid. Here, we identified the peak density of our exponential profile with the constant density of ref. 13, where a flattop density profile was assumed. Although, the model is valid for arbitrary profiles with a modified S parameter (see later). (b) If we still use the RES model with a lower density value (due to our exponential density fall) and so lower S parameter, the situation is more complex and probably the instantaneous “effective density” is relevant. In this case the compensation of the changing field amplitude by the corresponding effective density (higher amplitude compresses the plasma more leading to higher density) further suppresses this emission phase dependence as the referee also suggested. (c) However, even if we neglect this compensation, the intensity change within 2-3 pulses, i.e., 1-2 laser periods is maximum a factor of 2, which is a field amplitude as well as similarity parameter change of 1.41. This change in S (half step on the log scale of Fig. 5a,d in ref. 13) leads to an overestimated emission phase change of $2\pi/10$, which is negligible. (It is also small when comparing to our CEP measurement precision ± 210 mrad.)

A very recent paper: T. G. Blackburn et al., PRA 98, 023421 (2018) describes the RES model extended to a linear density ramp. Applying it to our case, we have an effective S parameter of 5.3-10.5 for the different scale lengths (in contrary to the former value of 67), which is still quite large and $S_{\text{eff}} \sim S^{0.5} \sim a_0^{-0.5}$, which leads to even less predicted change in the emission phase.

Certainly, introducing extra degree of freedoms in the form of different phase for the various APs an even better agreement can be obtained. However, as the determination of these extra 2 phases (for 3 APs) is a bit arbitrary, we stuck to the simpler and still satisfactory model of constant generation phase. We extended the text in the Methods to claim the constant emission phase assumption of our evaluation procedure and also softened the claim about CEP-dependence of the plasma surface motion as the parabolas are clearly CEP dependent, but a small effect of the phase might influence them that we didn't investigate.

I would also encourage them to present the actual dependence of the AP spacing retrieved in order that future theoretical works might be benchmarked against this data.

The AP spacing is determined by the Fourier transform (FT) of the CEP map (Fig. 2b). This FT map is shown below for a prepulse of 0.5 mm ($L/\lambda=0.13$). The signal amplitude in the different time ranges normalized differently for better visibility. The black lines signalize the peak at 3 and 5-6 fs, while their difference (dashed red line) gives the other peak around 3 fs.

As a comparison, the PIC simulation CEP map is shown below for ($L/\lambda=0.13$). The yellow lines signalize the CEP range, where the 5-6 fs peak is visible, i.e., 3 relevant attopulses exist.

3. As the laser pulse is short and the ions do not move significantly during the interaction, the plasma denting should be more or less symmetric, i.e. both forward and backwards motion of the plasma surface should be included. However, all the dependencies retrieved show only consecutive forward displacement of the points of XUV pulse generation. This is rather puzzling – can the authors explain it?

The plasma surface motion is quite symmetric in time that we describe with a parabola. However, we already observed earlier (ref [44]) that the attosecond pulse train for few-cycle lasers is typically asymmetrically generated relative to the laser envelope (see Fig. 2 in [44]) as well as to the parabolic motion of the plasma surface. Correspondingly, observed points on the parabolas are also asymmetric. In some parameter range they are shifted towards the rising edge, in other cases towards the trailing edge. Certainly, the plasma moves back at the trailing edge of the laser, but we do not have info in the presented cases about this motion as no intense attosecond pulses are generated in this time period

under our experimental conditions and in corresponding simulations. The figure below shows the electro-magnetic wave energy at higher frequencies ($50 > \text{harmonic order} > 10$) vs. plasma depth (x-axis) and time (y-axis) for the case of Fig. 4f ($\text{CEP} = 2\pi/3$, $L/l = 0.4$). The blue line indicates the spatio-temporal position of the peak of the envelope, while the purple lines parallel to it are the phases shifted by $\pi/2$ from the peak of the incident electric field, i.e., zero transitions from positive (electrons pushed in) to negative fields. The purple lines in a certain angle (relatively to a blue one) indicate the reflected electric field and along some of them we can see the generated attosecond pulses. The asymmetry is well visible between the attosecond pulse

train, the laser peak and the parabola.

Minor comments:

1. The title does not explicitly indicate that the results are related to high harmonic generation from a plasma surface. The “spectral interferometry” that is stressed instead does not seem to be the paper’s strongest quality to me, in terms of novelty and interest.

The spectral interferometry interpretation is applied the first time to HHG and it provides extraordinary results. We believe it will play an important role in the future. Therefore, we prefer to keep the original title with only a slight modification “Spectral interferometry with waveform-dependent relativistic high-order harmonics from plasma surfaces”.

2. I am not entirely sure that I completely understand the rationale behind using the term “waveform-dependent”. If the authors have a reason to differentiate “waveform-dependent” from “CEP-dependent”, this should be clarified, ideally in the abstract.

The two expressions are the same. We are using them alternately to avoid repeating the same expression.

3. It is worth mentioning in the main text (not in supplementary) that the plasma scale-length was linked to the prepulse setting by MEDUSA simulations, and a reference should be provided.

We have included MEDUSA in the main part not only in Methods.

4. The authors state that “each recorded XUV spectrum was tagged with the CEP of the corresponding laser pulse, measured with a single-shot stereo above-threshold-ionization (ATI) phasemeter (CEP-meter).” However, they also state that “the absolute CEP in experiments was determined by comparing simulations and experiments.” Thus it is not clear how they traced the CEP.

The CEP-meter gives a certain value that can be even calibrated to get the absolute CEP value (in the phase meter!). However, this is not the same value as in the experiment due to the different Gouy phase shift of the separated pulses (to the CEP-meter and the experiment). Sometimes also dispersive phase shift is important, if different amount of material is in the separated pulses. Therefore, we get the relative CEP for all shots in the experiment, but the absolute value are uncertain up to a constant. This constant is determined by comparing the measured CEP map to the simulated one. We included in the manuscript that it is the relative CEP that the phase meter provides.

5. When specifying the isolation degree, it is worth mentioning that it is for intensity, not for amplitude.

We included a short explanation in the manuscript.

6. On page 5 the authors write “Fig. 4f demonstrates such interpolation with similar parameters as in Fig. 4b.” Do the authors mean Fig. 4e in the latter reference?

Yes, Fig. 4e is meant. It has been corrected.

7. On page 7 the authors write “in this case reconstruction of the individual spectra of APs is not accurate, but it only leads to underestimation of the real isolation degree.” Why does this lead only to underestimation? It is not clear to me.

We use the measured spectrum, which has also noise. This noise leads to a background in the Fourier transform that we took (at corresponding temporal position, around 5-6 fs) as the signal (S_{p3}) for this case. The real S_{p3} signal in the “false time” domain might be smaller. The spectral intensity of the first (S_1) and last (S_3) APs are proportional and the middle (S_2) AP inversely proportional to this signal (see also Supplementary Fig. S1e). Therefore, overestimating S_{p3} means overestimating also S_1 and S_3 and underestimating S_2 . Correspondingly, we have similar over-/underestimations of the temporal intensities as well, which means we underestimate the S_2/S_1 and S_2/S_3 ratios, from which the smaller is the isolation degree.

A slight modification in the equation of the temporal coordinate of the evaluated parabola has been made, which now describes correctly 2D as well as boosted 1D geometry transformed back to the lab system. This did not influence the agreement between the simulation and experiment as the same

formerly old and now new equation is used for both of them. The new 55 deg angle of incidence and $L/\lambda=0.13$ scale length simulations in Fig. 5 have a stronger influence, but still provide good agreement.

We hope our answers and revision of the manuscript is satisfactory and the reviewer finds our paper suitable for acceptance.

Reviewer #3

We thank the referee for the review and appreciating our results. We revised the manuscript implementing the questions and concerns of the referee. In the answers we used the reference numbers of the original (not resubmitted) manuscript.

1. through the paper, different plasma scale lengths are used in the simulation than the experimental ones obtained from the MEDUSA code given a certain pre-pulse delay. Is there some physical reason why the authors chose a different parameter or it is just to fit the experiment? This should be clarified. In particular, for the $L/\lambda = 0.13$ case, the simulations in figure 2 have been done at $L/\lambda = 0.1$ while in figure 5 they have been done at $L/\lambda = 0.15$. While I can understand that there could be a disagreement between the scale length derived with the MEDUSA code and the one used in the simulations, I cannot understand why choosing a different theoretical number (0.1 and 0.15) for the two cases. For the same reason, I would not say that “Both effects.. are in perfect qualitative agreement with the experiment” as stated at the end of the “Plasma surface denting” section. It would be useful to show how much the theoretical results change when you use the L/λ parameter retrieved from the

experimental pre-pulse delay and compare it with the one that fits the data, in order to let the reader understand what is the order of magnitude of the disagreement if there is any.

The original simulations were made at different scale length values with a step of 0.05λ and we picked the two simulations that agreed better with the experimental results in Fig 2 and 5. Now, we performed (a lot of) new simulations with 55° angle of incidence and more scale length values as shown in the side figure (including the measured result). It shows the dependence of the CEP map on the scale length. We compare its Fourier transform (FT) to the FT of the measurement as the 5-6 fs peak dependence on the CEP was found to be the best indication of their agreement and the absolute CEP definition in the measurement. The value of 0.13 seems to agree the best with experimental CEP map (Comparison1) and simultaneously with corresponding parabolas (Comparison2). Even though slightly different scale lengths may agree better for one of the two comparisons, they lead to significant disagreement for another one.

We modified our text to: “Both effects – significant dependence on CEP and plasma scale length – are in qualitative agreement with the experiment. The differences between experiment and 1D simulations might originate from multi-dimensional effects as in the λ^3 -regime [Naumova PRL 92, 063902 (2004)].”

2a. In the “Spectral Interference” section the authors assume that the spectral phase of the first AP is flat and so they claim they can measure the phase of the second pulse.

We should clarify this point. Spectral interferometry measures the phase difference between the first and second pulses. Based on simulations with different CEPs we observed that under our experimental conditions none of the pulses have significant 2nd or higher order spectral phase, i.e., they are Fourier limited (see new supplementary Fig. S2). Therefore, we can assume that in the experiment one of the pulses, for example the first, has flat phase (no higher order spectral phase) and consequently the measured difference is the phase of the second attopulse. Certainly, we could also assume that the second pulse has flat phase and correspondingly calculate the phase of the first from the measured difference.

This spectral interference measurement would be useless if the error in the spectral phase difference is so large that the above calculated phase of the other (second) pulse with its spectrum would provide a significantly longer pulse than the Fourier limit. In other words, it would not be clear if this elongated pulse is real or originating from the measurement error. Especially, it would not agree with the simulations, where all pulses were chirp free.

Therefore, the quality of the measurement is reflected by the fact that the Fourier transform of the above determined phase with its spectrum gives a practically also Fourier limited pulse as shown by the red line in Fig. 3d. We checked this property for multiple measurement results and are confident to claim that the measured phase difference is consistent with Fourier limited attopulses.

To emphasize this “freedom” of choosing the pulse for which we assume the flat phase, we modified the text to “Additionally assuming the spectral phase of one of the APs (for example 1st) is flat, ..., the phase of the other one (2nd) becomes equal to their difference”.

They refer to previous simulations from Ma G et al. [Phys. Plasmas 22, 24879 (2013)] where no clear information on the second and third order of the phase of the generated pulses seems to be given. The flat phase can be argued from FIG.1 of that paper, where the attosecond burst seems to have the same delay for every photon energy, but it is difficult to estimate second and third order terms from this graph (and in any case FIG.1 refers to a different CEP value).

We inserted a new figure (Fig. S2), where for one dominant attopulse the spectral phase difference minus the linear term from the evaluation of a PIC simulation spectrum and the spectral phase of the individual pulses are plotted indicating that they are chirp free. We also added other simulations (in Fig. S3) showing the same flat spectral phase for two attopulses with similar intensity (corresponding to a different laser CEP) as well as different scale length values.

We also inserted a short text in the caption claiming that the attopulses are chirp free for all laser CEPs as we observed it in simulations.

It's not clear why we could not assume that the phase of the second pulse is flat and so retrieve the phase of the first one. Are second and third order terms greater in the second pulse? Is this supported by theoretical simulations previously done?

As described above, we can also assume that the phase of the second pulse is flat and so retrieve the phase of the first one. We again obtain that the first is practically Fourier limited. The 2nd, 3rd, and higher order spectral phase terms are practically zero for all attopulses in the simulations in good agreement with the measured phase differences within measurement error.

2b. Dropping the assumption that the phase of the first pulse is flat, the only thing that can be measured, as the authors state, is the difference of phase between the two pulses. This difference is shown in figure 3c. However, this phase is not easy to read. The linear dependence must be subtracted in order to show the differences in the second and third order terms of the two pulses. Even if the duration is not affected so much, as shown in figure 3d, some wings appear in the pulse, which suggests the presence of higher order terms that cannot be appreciated from figure 3c if the linear phase is not subtracted. A comparison of this second and third order terms with theoretical predictions would add value to this section, focusing more on what can be actually measured.

We agree with the referee that the spectral phase without the linear dependence is more relevant.

Therefore, we replaced the phase with this difference in Fig. 3c and 4d. They show a very small deviation from the horizontal line indicating a phase flat enough. We inserted an extended figure here in this answer with the derivatives of this phase as well, the group delay and the GDD. They also show quite flat behavior. Furthermore, they reflect that the observed deviation from the Fourier limit in the time domain originates from a spectral range with low spectral intensity (52-57 eV), where the noise was higher (no metal filter was used during the measurement!). As the GD and GDD curves fluctuate around zero, therefore their effect cannot be estimated from this plot. However, the temporal FT of the spectral amplitude with the obtained spectral phase plotted in red in (d) evaluates their effect well. Correspondingly, the measurement result as well as the theoretical (numerical) prediction show a flat phase, and so no 2nd, and 3rd order spectral phase that we emphasized in the text.

3. In the section “Plasma surface denting”, assuming that one attosecond pulse has a flat-phase, the authors can retrieve the phase of the other two pulses showing that they also have a linear phase. Again, no clear reference is given on this assumption. As a consequence, some sentences should be rephrased, like “These observations indicate that all generated APs in the measurement were chirp-free”.

This flat-phase property is very important for our work as it permits to obtain various extra information and we have observed it in our simulations in a broad parameter range. We inserted Fig. S2 that shows the flat phase for the case of a dominant and two small attopulses. We also inserted Fig. S3 indicating the flat phase for different CEP and scale length values. Therefore, the simulations predict chirp-free attopulses in the relevant parameter range, which is not an assumption. It is the motivation to assume a similar flat phase in the measurements. We changed the sentence to “These observations are consistent with the numerical predictions that all generated APs in the measurement were (atto-)chirp-free”.

minor issues:

1. please give a definition of the normalized vector potential, it could be useful for a broad audience

We inserted the equation in the text.

2. I do not understand why there are different CEPs values in the text and in the captions. For example, in the “spectral interference” section, $\Phi_{\text{CEP}} = 0$ according to the text, while it is $\pi/3$ according to the caption. Also in the “plasma surface denting section”, $\Phi_{\text{CEP}} = \pi$ in the text, while it is equal to $2\pi/3$ in the caption.

We thank the referee recognizing this mistake. We corrected this by writing: “According to our simulations, for $\varphi_{\text{CEP}} \approx 0$ (and $L/\lambda=0.25$) two almost equivalent APs are produced,...”. We also matched the caption with $\varphi_{\text{CEP}} \approx 0$. We also corrected the sentence “In the $\varphi_{\text{CEP}} \approx \pi$ case (and $L/\lambda=0.4$),...” and matched the caption with with $\varphi_{\text{CEP}} \approx \pi$.

A slight modification in the equation of the temporal coordinate of the evaluated parabola has been made, which now describes correctly 2D as well as boosted 1D geometry transformed back to the lab system. This did not influence the agreement between the simulation and experiment as the same formerly old and now new equation is used for both of them. The new 55 deg angle of incidence and $L/\lambda=0.13$ scale length simulations in Fig. 5 have a stronger influence, but still provide good agreement.

We hope our answers and revision of the manuscript is satisfactory and the reviewer finds our paper suitable for acceptance.

Reviewers' comments:

Reviewer #1 (Remarks to the Author):

The authors have addressed all the issues I raised in the first review. I appreciate their efforts showing new simulation results to support the experimental results. This article should be suitable for publication in Nature Communication at present.

Reviewer #2 (Remarks to the Author):

The authors convincingly answered to all my comments and incorporated appropriate clarifications/corrections into the text. I can recommend the current version of the manuscript for publication in Nature Communications.

Reviewer #3 (Remarks to the Author):

I thank the authors for the detailed answer. The revised manuscript and the response to the referees' letter answered many of the concerns that I had. The addition of Fig. S2 and the new simulations reported in Fig. S3 exhaustively confirm that the generated pulses have flat spectral phases, which is the main property that allows applying the interferometric approach. The new simulations shown in figure 2 and 5 are now consistent with the scale length estimated with the MEDUSA code. There's one last comment that I have about these last results.

It seems now that the agreement between simulation and experiment in figure 2 is worsened. The reasonable agreement claimed in the corresponding section (first two lines, page 3) should be contextualized. In the "Response to Referees Letter", the authors state that: "the value of 0.13 seems to agree the best with experimental CEP map (Comparison1) and simultaneously with corresponding parabolas (Comparison2)." I think that this should be explicitly said (somewhere) in the paper since now it's not evident the agreement between figures 2b and 2c. Moreover, I cannot understand how the absolute CEP was determined by comparing these two maps since they are very different. In particular, the continuous shift of the harmonics is faster in the theoretical results than in the experimental results. Moreover, the theoretical result shows jumps that are not experimentally observed. If the absolute CEP was calibrated with some procedure that gets rid of the disagreements observed (maybe comparing the Fourier transform of the map) this should be said. If also the absolute CEP was calibrated by looking at the parabolas, this should be specified.

Apart from these last remarks, I think that the new version is suitable for publication in Nature Communications.

Answer to Reviewer #3:

I thank the authors for the detailed answer. The revised manuscript and the response to the referees' letter answered many of the concerns that I had. The addition of Fig. S2 and the new simulations reported in Fig. S3 exhaustively confirm that the generated pulses have flat spectral phases, which is the main property that allows applying the interferometric approach. The new simulations shown in figure 2 and 5 are now consistent with the scale length estimated with the MEDUSA code. There's one last comment that I have about these last results.

We thank the referee for the very practical suggestions! We extended the text.

It seems now that the agreement between simulation and experiment in figure 2 is worsened. The reasonable agreement claimed in the corresponding section (first two lines, page 3) should be contextualized. In the "Response to Referees Letter", the authors state that: "the value of 0.13 seems to agree the best with experimental CEP map (Comparison1) and simultaneously with corresponding parabolas (Comparison2)." I think that this should be explicitly said (somewhere) in the paper since now it's not evident the agreement between figures 2b and 2c.

We extended the text around Fig. 2 c as well as Fig. 5:

"Fig. 2c shows the particular case $L/\lambda_L = 0.13$, which is in reasonable agreement simultaneously with the experimental data in Fig. 2b as well as the later evaluation in Fig. 5. Supplementary Fig. S1 shows the simulated CEP-sorted spectra for different scale lengths and indicates that if only the results in Fig 2 are compared than $L/\lambda_L = 0.1$ shows a better agreement. The absolute CEP in experiments was determined by comparing simulations and experiment (see Methods)."

Furthermore, we inserted a new supplementary figure that shows the simulated CEP maps for different scale length.

After Fig. 5:

"This agreement together with the visual comparison between Fig. 2b and c were the criteria to choose the scale length in simulation."

Moreover, I cannot understand how the absolute CEP was determined by comparing these two maps since they are very different. In particular, the continuous shift of the harmonics is faster in the theoretical results than in the experimental results. Moreover, the theoretical result shows jumps that are not experimentally observed. If the absolute CEP was calibrated with some procedure that gets rid of the disagreements observed (maybe comparing the Fourier transform of the map) this should be said. If also the absolute CEP was calibrated by looking at the parabolas, this should be specified.

The Fourier transform of the CEP maps were used to define the absolute experimental CEP value. The Fourier transform (FT) map of Fig 2b, i.e., a prepulse of 0.5 mm ($L/\lambda=0.13$), is shown below. The signal amplitude in the different time ranges normalized differently for better visibility. The black lines signalize the peak at 3 and 5-6 fs, while their difference (dashed red line) gives the other peak around 3 fs.

As a comparison, the FT CEP map of the PIC simulation from Fig. 2c is shown below, i.e., for ($L/\lambda=0.13$). The yellow lines signalize the CEP range, where the 5-6 fs peak is visible, i.e., 3 relevant attopulses exist.

We defined the absolute CEP in the experiment based on this range.

We describe in the methods with an extra sentence:

“The absolute CEP on the target was obtained from the relative values by comparing the Fourier transform of the experimental CEP-sorted spectra with that from PIC simulations so that the 3rd AP indicated by the 5-6 fs peak appears in the same CEP range.”

Apart from these last remarks, I think that the new version is suitable for publication in Nature Communications.

We thank the referee for the positive feedback and evaluation!

REVIEWERS' COMMENTS:

Reviewer #3 (Remarks to the Author):

The authors have convincingly modified their manuscript in response to my comments and I have no further comments to add. I recommend publication of the current manuscript in Nature Communications.